# The Mediating Effect of Functional Health on the Relationship between Social Capital and Cognition among Chinese Older Adults

**DOI:** 10.3390/ijerph192316123

**Published:** 2022-12-02

**Authors:** Xinyu Liu, Shuangshuang Wang, Siqi Liu, Nengliang Yao, Quan Wang, Xiaojie Sun

**Affiliations:** 1Centre for Health Management and Policy Research, School of Public Health, Cheeloo College of Medicine, Shandong University, Jinan 250012, China; 2NHC Key Lab of Health Economics and Policy Research, Shandong University, Jinan 250012, China; 3School of Public Administration, Southwest Jiaotong University, No. III, North Section I Second Ring Road, Chengdu 610031, China; 4Center of Health System and Policy, Institute of Medical Information & Library, Chinese Academy of Medical Sciences & Peking Union Medical College, Beijing 100020, China; 5Home Centered Care Institute, 1900 East Golf Road, Suite 480, Schaumburg, IL 60173, USA; 6School of Public Health, Peking University, Beijing 100871, China; 7Brown School, Washington University in St. Louis, St. Louis, MO 63130, USA

**Keywords:** social capital, cognition, functional health, mediating effect

## Abstract

This study evaluated the association between cognition and social capital among Chinese older adults and analyzed the mediating role of functional health in that interaction. The baseline and follow-up data were acquired from the 2011 and 2015 China Health and Retirement Longitudinal Study (CHARLS). The sample included 6291 adults aged 55 years and above in 2011. The dimensions of social capital included social trust, social support, social participation, and reciprocity. Cognition was measured based on mental state and episodic memory. The Karlson–Holm–Breen method was employed to explore the association between social capital at the baseline and cognition four years later, and the mediating role of baseline functional health. There was a positive relationship between financial support (one of the social support variables) and reciprocity, and mental state (β = 0.186, *p* = 0.00; β = 0.306, *p* = 0.012). Furthermore, a positive relationship between social participation and episodic memory (β = 0.129, *p* = 0.002) was observed. The mediating effect of functional health explained 7.7% of the total effect of social participation on episodic memory. These findings may contribute to research concerning the potential explanation of the association between social capital and cognition as well as interventions aimed at improving cognitive performance in older adults.

## 1. Introduction

As the proportion of older adults rapidly increases in China, the economic status and social role of this population face dramatic changes, the most prominent being the growing number of elderly patients with declining cognition, as demonstrated by an increase in dementia cases [1]. A previous study reported that Chinese people with dementia accounted for nearly one-fourth of the total number of people with dementia worldwide [2]. Therefore, it is essential to explore factors affecting cognition in older adults.

### 1.1. Cognition

Cognition refers to a psychological process crucial to activities of daily living, such as understanding, perception, memory, imagination, and thinking [3]. Cognition can be affected by genetic, lifestyle, and social-psychological factors [1]. Cognitive ability is composed of crystallized intelligence and fluid intelligence [4]. Crystallized intelligence refers to the knowledge and skills accumulated by individuals through education or training, and fluid intelligence refers to the performance of individuals in learning and processing new information [5]. The existing literature usually adopts episodic memory ability (e.g., date cognition, calculation, and drawing) and mental state (e.g., phrase recall test) as representatives of fluid intelligence and crystallized intelligence when measuring individual cognitive ability [6].

### 1.2. Social Capital

The positive role of social capital in cognition has been gradually confirmed [7]. From the communitarianism perspective, Putnam defined social capital as a social network based on mutual trust and reciprocity that encouraged individuals to participate in collective activities and acquire resources and support [8]. According to this definition, social trust, social support, social participation, and reciprocity can be regarded as key indicators of various dimensions of social capital [9]. Social trust emphasizes mutual trust between different individuals [10]; social support refers to emotional and financial support [9], which is an individual’s subjective perception of the availability of external assistance or support from their social network [11]; social participation includes participation in cultural and recreational activities or parties [6]; and reciprocity indicates the willingness or capacity to help others and attain help from others [9]. Some studies have highlighted that multiple dimensions of social capital may be consequential for cognition, and the association between social capital and cognition may vary with the divergence of social capital dimensions and cognition types [7].

### 1.3. Social Capital and Cognition

Existing studies on older adults focus on the following two dimensions of social capital: social participation and social support. Regarding social participation, Chinese scholars have found that lower levels of social participation could be a risk factor for cognitive impairment among older adults [1], and engagement in social activity could protect against age-related cognitive decline [7]. For example, playing cards and mahjong is associated with better cognition [12]. Kelly et al. also reported similar results in a systematic review [7] of the relationship between social capital and cognitive function. The review included 39 studies in different countries and demonstrated a relationship between social activity and cognition globally. 

The existing literature on social participation has revealed complicated effects on the different domains of cognition. Mortimer [13] and James [14] found that social participation could improve both mental status (i.e., language fluency and visual space) and episodic memory. However, several other studies indicate that social participation has no significant effect on mental status or episodic memory [15]. The study conducted by Bosma [16] underlined that social participation exclusively improved episodic memory but had no significant effect on mental state [16]. Zhu and Zeng [6] and Mousavi-Nasab [17] found that social participation had a more significant effect on the episodic memory of older adults than on their mental state. Very few studies have indicated that social participation has a significant negative effect on mental status but a positive effect on memory [18]. For instance, Meister [19] found that social participation was related only to mental state and not episodic memory. The inconsistencies among the studies may be due to differences in the age composition of samples, measurement tools of the research variables, and environment of the sampling areas. Another possible reason is that different studies have different control criteria for the level of significance.

Regarding social support, Andrew and Rockwood [20] found that, compared to their counterparts with higher social support, older adults with lower social support might face an approximately 40% increase in the odds of cognitive decline. Ellwardt [21] confirmed that emotional support, as a key aspect of social support, was predictive of cognitive improvement among older adults.

Some studies have found that different types of social support may cause varied effects on the same cognitive dimension, while the same type of social support has different effects on the different dimensions of cognition (i.e., episodic memory ability and mental state). Ellwardt [21] found that emotional support was more closely associated with cognition improvement than financial support. Hughes [22] found that social support was only related to the improvement of mental status and not episodic memory.

The literature shows that the influence of different dimensions of social capital on cognition may be multifaceted and varied, and hence worthy of further analysis. To remedy the knowledge gap, a comprehensive framework of social capital indicators is necessary to go beyond social support and social participation and explore the relationship between different dimensions of social capital and different cognitive capabilities. 

### 1.4. Current Study

The present study aims to address the limitations of the previous studies in two aspects. First, most researchers have focused on the association between one exclusive social capital dimension, such as social support or social participation, and cognition, while not fully considering the comprehensive scope of social capital, and dismissing the roles of social trust and reciprocity [23]. While some studies constructed robust measures of social capital, the multiple roles played by different dimensions of social capital in the performance of the two cognitive domains (i.e., episodic memory and mental state) have not been fully evaluated. Second, most of the related studies have explored the relationship between a specific dimension of social capital and cognition among older adults based on a relatively small-scale sample within one province or several cities, creating the need for stronger evidence based on a larger national-level population sample. Third, the empirical studies on the relationship between social capital and cognition are yet to explore whether they are related or causal, calling for a discussion on the potential mechanisms between them. 

### 1.5. Functional Health as a Mediator between Social Capital and Cognition

Functional health may partly explain the relationship between social capital and cognition [24]. Several studies have suggested that older adults with a higher level of social capital may undertake more social roles, have a greater sense of social participation, and have more outside resources and information, all of which are conducive to the maintenance of functional health [25]. Furthermore, functional health is an important factor affecting cognition. Older adults with excellent functional health may have more advantages in communication, information flow, and problem-solving skills, which could effectively expand their social circles and maintain their cognitive ability [6]. Moreover, as discussed above, social capital has been proven to be a significant factor affecting cognition [26]. Since social capital is an influencing factor for both physical and cognitive functions, scholars have also studied the relationships between social capital, physical function, and cognitive function. It has been revealed that cognitive ability plays an intermediary role between social capital and physical functional health [26]. The decline of functional health leads to a decrease in brain activity in older adults [27]; simultaneously, the decline in cognitive function could lead to a decline in the ability of daily living (ADL). Therefore, if the ability of daily living and cognitive ability are interrelated, it is necessary to determine the working path from one side to the other. The existing studies have shown that cognitive ability mediates the relationship between social capital and physical health; however, whether physical health also plays an indirect role between social capital and cognitive ability remains unclear. 

### 1.6. This Study

This study aims to (1) measure social capital based on the four dimensions of social trust, social participation, social support, and reciprocity; (2) use nationally representative data to test the associations between the different types of social capital and cognitive domains; and (3) analyze the extent to which functional health plays a mediating role in the relationship between social capital and cognition. The conceptual framework of this study is shown in Figure 1. 

## 2. Methods

### Sample and Procedure

The data were acquired from the China Health and Retirement Longitudinal Study (CHARLS). Community-dwelling adults aged 45 years and above were selected using a multistage, stratified probability sampling design from 450 villages and 150 counties in 28 provinces in China (http://charls.pku.edu.cn/) (accessed on 15 January 2020). The survey began in 2011 and follow-up interviews were conducted biennially thereafter. All the participants signed an informed consent form prior to the data collection. The data applied in this study and other detailed information on CHARLS are available online (http://charls.pku.edu.cn/) (accessed on 15 January 2020). 

This study used data from the 2011 and 2015 releases of CHARLS. A total of 6291 individuals were selected according to the following inclusion criteria: (1) aged 55 years and older as of 2011, since people over 55 years old had already begun to show mild cognitive impairment (MCI) [28]; (2) did not display signs of cognitive impairment in 2011; (3) had valid information on cognition, social capital, functional health, and control variables in 2011; and (4) had valid information on cognition in 2015. The detailed sampling process is illustrated in Figure 2.

## 3. Measurement

### 3.1. Cognition

Cognition was measured using the Chinese version of the Mini-Mental State Examination (MMSE), which includes the perspective of two domains, that is, mental state and episodic memory [6]. Mental state, as a typical representative of this solidified ability, includes tests of date cognition, calculation, and drawing. These three methods can comprehensively reflect the solidified ability of knowledge and skills acquired by individuals through education and training. For date cognition, the respondents reported the year, month, day, season, and day of the week in which they were interviewed. One point was given for each item that was correctly answered, with the scores ranging from 0 to 5. The calculation test asked the respondents to count down from 100, subtracting seven each time for five consecutive times. The respondents received one point each time they calculated the number correctly. The scores ranged from 0 to 5. For the drawing test, the respondents were asked to draw a picture displayed by the interviewer; those who drew the picture successfully received 1 point. The final mental state score was the sum of the three tests and ranged from 0 to 11, with higher scores reflecting a better mental state. Episodic memory was measured using a phrase-recall test. Interviewers read ten words to the respondents and then asked them to recall the words. One point was assigned for each correct recall. The score of the phrase recall ranged from 0 to 10, with the higher scores indicating better episodic memory. These cognition measurement methods demonstrated good internal reliability (Cronbach’s alpha = 0.898). Cognition scores were calculated for the 2011 and 2015 datasets.

### 3.2. Social Capital

According to Putnam’s social capital theory [29], the measurement of social capital usually includes four dimensions: social trust, social support, social participation, and reciprocity. For social trust, we used the respondents’ places of residence as the variable. Respondents living in the village/community they were born in were defined as having “high” social trust, and those living in a place different from their birthplace were categorized as having “low” social trust [9]. Using the birth and living places as a measure to assess social trust has appeared in previous studies [9], as individuals living in their place of birth tend to trust their neighborhoods more and are most closely connected [30]. In rural China, under the influence of traditional culture, when the birth and living places of the respondents are the same, it implies that most of the respondents’ neighbors have the same family name and are therefore more closely connected and tend to trust one another [31]. Similarly, in urban areas, older adults living in their place of birth usually form a social network of acquaintances, including their friends, colleagues, and neighbors. This type of network ensures close communication and strong emotional ties, maintaining a strong level of trust between them [32]. Therefore, birthplace could be an appropriate variable for measuring social trust in China [31]. 

Social support was measured in terms of financial support and care. The respondents were asked whether they had received financial support from their children in the past year. They answered either “yes” or “no”. Care was assessed by asking the respondents “Do relatives (except spouse) or friends provide long-term care when you need it?” We defined care as “yes” if the answer was “yes, they do”, and “no” if the answer was “no, they don’t” [33]. Regarding social participation and reciprocity, the respondents were asked whether they had participated in the following eight types of activities in the last month: (1) interacting with friends; (2) playing mahjong, chess, cards, or going to other community clubs; (3) providing help for free to relatives, friends, or neighbors who did not live with the respondent; (4) going to sports, social, or other clubs; (5) taking part in community-related organizations; (6) undertaking voluntary or charity work; (7) providing help for free to patients or disabled people who did not live with the respondent; and (8) going to school or attending training courses. If the respondents answered “yes” to items (1), (2), (4), (5), or (8), social participation was defined as “yes”, and “no” if they did not participate in any of the five activities. Reciprocity was assessed using the answers to questions (3), (6), and (7). If the respondents answered “yes” to any of them, reciprocity was defined as “yes”, or “no” if they answered “no” to all the activities of (3), (6), and (7) [9]. Reciprocity, as measured in this study, may be seen as an asymmetric interpersonal relationship and its return unpredictable. This reciprocal behavior follows the dual logic of “supporting the weak” and “reciprocity”, both of which can be integrated by maximizing group utility with altruism as the core. Despite this asymmetry, it remains a symbol of positive interactions between individuals and groups. Studies have shown that such a relationship could also have a positive effect on health [26,34]. Scholars at home and abroad have used similar methods to measure reciprocity [9,35]. 

### 3.3. Functional Health

Functional health was measured using activities of daily life (ADL), including dressing, bathing, eating, getting on and off the bed, going to the toilet, and controlling bowel movements), and instrumental activities of daily living (IADL), including doing housework, cooking, shopping, managing money, and taking medicine. Both are commonly used tools for screening poor functional health [36]. The respondents were asked if they experienced any difficulty completing these activities; four possible answers for all activities emerged: “no difficulties”, “has difficulties but achievable”, “has difficulties and needs help”, and “unachievable”. If the participants reported any difficulty with ADL and IADL (the last three options), it was defined as an unhealthy function. If the participants reported no difficulty with ADL and IADL (the first option), it was defined as a healthy function. Thus, functional health was defined as a binary variable.

### 3.4. Control Variables

The control variables were drawn from the previous studies on the cognition of older adults, including demographics, socioeconomic status, and health-related variables [12,37]. The demographic variables included age (55–59 years, 60–74 years, ≥75 years), gender, and marital status (married, divorced, widowed, or unmarried). The indicators of socioeconomic status included having money in the bank (yes or no), educational level (illiterate, primary school or lower, secondary education, or higher education) [9], and *hukou* status. *Hukou* refers to household or residential administration regulations and includes rural *hukou*, non-rural *hukou* and unified *hukou* (the unified format of rural and urban residents’ household registration). Self-rated health conditions and sleep were used as health-related variables. The respondents were asked to rate their health on a five-point scale ranging from 1 = very good to 5 = very poor. The answers were recorded as good (very good/good), fair (fair), or bad (poor/very poor) [9]. Sleep was assessed by the average hours of sleep per night, and, according to a related study, the sleep time at night was divided into six periods (≤5 h, 6 h, 7 h, 8 h, 9 h, ≥10 h) [38]. The baseline cognition was also included as a control variable.

The data for social capital, functional health, and the control variables were collected from the 2011 baseline survey, and the data for the dependent variables (cognition in 2015) were collected from the results of the 2015 survey. 

### 3.5. Statistical Analysis

First, the demographic characteristics using the mean (standard deviation) or frequencies (percentages) were depicted. Second, Spearman’s coefficients were used to test the correlation between the cognitive variables; t-tests and an ANOVA analysis or Kruskal–Wallis equality of populations rank test were used to compare the cognitive scores across different subgroups of older adults. If the Bartlett test for equal variances among different groups was not statistically significant, an ANOVA analysis was used to test the differences of the means; otherwise, the Kruskal–Wallis equality of populations rank test was used (See Table 1). 

Third, to test the mediation effect of functional health on the association between the different dimensions of social capital and cognition, we conducted a linear regression model, based on the Karlson–Holm–Breen (KHB) method [39]. Since the dependent variable (i.e., cognition) in this study was binary, the use of the KHB coefficient decomposition method became necessary, as it could effectively avoid the error caused by directly comparing the coefficients of a logistic nested model. The KHB decomposition method has been widely used to solve the scale problem in logistic regression models [26].

Taking the different dimensions of social capital as independent variables, this study conducted a linear regression analysis for each cognitive result while controlling for the control variables. As shown in Figure 1, the total effect is the effect of social capital on cognition. The direct effect is that of social capital on cognition when controlling for functional health. The indirect effect is the effect of social capital on cognition through functional health. The proportion of the mediating effect to the total effect (i.e., the magnitude of the mediating effect) is calculated by dividing the indirect effect by the total effect. In the KHB analyses, the social capital indicators were social trust, financial support, care, social participation, and reciprocity, and cognition was mental state or episodic memory. All the analyses were adjusted for the control variables. All the statistical analyses were performed using STATA version 15.0 (Stata Corporation, College Station, TX, USA). The statistical significance was defined as a two-tailed *p* value less than 0.05.

## 4. Results

### 4.1. Baseline Participant Characteristics 

This study included 6291 participants; their baseline characteristics are presented in Table 1. For the cognition variables, the mean mental state score decreased from 7.1 in 2011 to 6.6 in 2015, and the mean episodic memory score decreased from 3.5 in 2011 to 3.4 in 2015. The older adults with higher mental state and episodic memory scores were generally younger, male, married, had money in the bank and a higher education level, had a non-rural *hukou*, better self-rated health and healthier functional health, and slept about 6–7 h at night. 

### 4.2. The Associations of Different Dimensions of Social Capital with Cognition

As shown in Table 2, after adjusting for the control variables, financial support was significantly associated with a high mental state score (β = 0.19, *p* = 0.009), and the older adults with higher levels of financial support were more likely to have a higher mental state score. Reciprocity was significantly associated with high mental state scores (β = 0.31, *p* = 0.012), and the older adults with higher levels of reciprocity were more likely to have higher mental state scores. However, no significant relationships were found between the other social capital dimensions and mental state (*p* > 0.05) (Table 2). 

As shown in Table 3, social participation was significantly associated with high episodic memory scores (β = 0.13, *p* = 0.002), and the older adults with higher levels of social participation were more likely to have a higher episodic memory score. However, this study did not reveal significant relationships between any other social capital dimensions and episodic memory (*p* > 0.05) (Table 3).

### 4.3. The Mediating Effect of Functional Health 

To control for potential confounders at the baseline, the KHB was used to verify the mediation effect. We found that the direct, indirect, and total effects were statistically significant when the independent variable was social participation, and the dependent variable was episodic memory (see Table 3 and Figure 3). Therefore, it could be concluded that functional health played a partial mediating effect on the association between social participation and episodic memory, and the mediating effect of functional health explained 7.7% of the total effect of social participation on episodic memory (*p* = 0.021) (Table 3).

## 5. Discussion

Cognition is critical to the quality of life of older adults. Therefore, it is imperative to study and understand cognition to prevent and treat cognitive impairments. Although previous studies showed that some dimensions of social capital could protect cognition, and functional health could be affected by social capital and closely related to cognition, it remained ambiguous whether and to what extent functional health played a mediating role in the relationship between social capital and cognition. The results indicated that financial support and reciprocity were associated with a better mental state, and social participation was associated with better episodic memory. A further mediation analysis found that functional health did not mediate the association between financial support or reciprocity and mental state, whereas the association between social participation and episodic memory was partially mediated by functional health among older adults. 

The previous studies demonstrated that financial support and reciprocity were associated with better cognition, as well [26]. The scholars believed that financial support could not only improve living conditions but also maintain older adults’ sense of social exchange and self-esteem [40], while reciprocity could relieve the psychological pressure, which was helpful for cognition. 

Additionally, we found that financial support and reciprocity were more likely to benefit the mental state of the older adults than their episodic memories. The study participants were distributed in rural areas with relatively low socioeconomic status (79% of rural household registration), and the elderly in rural areas could not meet their healthcare needs owing to the limited healthcare resources and heavy financial burden in rural areas. Adequate economic support can relieve the living pressure and reduce the mental burden; therefore, economic support significantly affects the mental state of the elderly. Regarding reciprocity, in traditional Chinese culture, the spirit of “keeping watch and helping each other” has always been regarded as a kind of virtue. Therefore, the daily mutual assistance and support between neighbors has become a social expectation. Even under rapid economic and social changes, this tradition of mutual assistance continues to be practiced in vast rural areas of China. Mutual help from family, friends, and neighbors enhances the self-health management ability and long-term maintenance of the elderly, contributing to their physical and mental development. Meanwhile, compared with episodic memory, the individual mental state depends more on the long-term accumulation of material and human capital [41]. Therefore, the mental state may be more sensitive to investment in financial support and interpersonal assistance. 

Furthermore, we found that social participation was not significantly related to mental state. This implies that the role played by social participation is not only related to the existence of social participation but also to the autonomy of social participation. An existing study showed that social participation might not always benefit the health of the elderly. Participation in society on an autonomous basis benefits their mental state or mental health; however, on a mandatory basis, its impact may be negative. The positive and negative effects of social participation on mental state may offset each other; therefore, we did not observe any significant impact [42]. Simultaneously, we found a correlation between social participation and episodic memory. Schwartzman et al. found that the ability of situational memory changed greatly in the life cycle and might be affected by other factors, especially social activities [43].

Our study suggests that functional health mediates the association between social participation and episodic memory. Social participation can promote physical and mental activities in older adults. Physical activities can improve functional health, whereas mental activities can reduce memory decline. Physical activity in older adults can improve their cardiopulmonary and cerebrovascular function, prevent neuropathological changes, and help delay cognitive decline. Additionally, older adults often need to communicate with others in social activities, improve their vigilance, and observe and participate in their surroundings simultaneously. The repetition of these cognitive activities may be related to memory maintenance in older adults [6]. However, the mediating effect of functional health on the total effect of social participation on episodic memory was relatively low. This may be because social participation affects memory function more through mental health and other pathways.

Our study did not find any significant associations between social trust, receiving care, and cognition. The study of Zhang et al. [44] suggested that older adults who often received care could feel burdened or guilty, leading to psychological distress that negatively affects their cognitive health. However, another study indicated that receiving care from children increased access to health information and enhanced the spiritual comfort of older adults [33], which could benefit their cognition. The positive and negative effects of receiving care might offset each other; therefore, we did not observe any statistically significant results. In addition, using residence as a proxy variable of social trust, while birthplace is endogenous when it is associated with other social and health characteristics, may lead to the non-significance of social trust measured in this analysis [10]. 

## 6. Limitations and Suggestions

This study had some limitations. First, although the methods of measuring social capital indicators in this study have been applied by other Chinese scholars [9], knowledge about the validity of these social capital measures remains restricted owing to the characteristics of the secondary data and binary variables used in the past. For example, the measures did not take the frequency or type of social activities into account [7]. In addition, residence was regarded as a proxy variable of social trust, while birthplace is endogenous when it is associated with other social and health characteristics, which may render the social trust measured in this analysis non-significant. Future studies should use more comprehensive assessment tools. Second, owing to data limitations, we verified only one mediation variable in this study; therefore, the proportion of the mediating effect to the total effect was relatively lower. In the future, it will be crucial to explore the potential mechanisms underlying the association between social capital and cognition among older adults. Despite these limitations, this study contributes to the current literature regarding the cognition of older adults through its large national-level sample, longitudinal data, and multidimensional construction of social capital. 

## 7. Conclusions

This study demonstrates that social capital dimensions such as financial support, reciprocity, and social participation are associated with cognition. Moreover, the association between social participation and episodic memory is partially mediated by functional health among older Chinese adults. However, the mediating effect of functional health constitutes only a small proportion of the total effect, and further studies are required to examine the mechanisms underlying this association in more detail. Our study suggests that primary healthcare providers should pay more attention to older adults who lack financial support, social interaction, and social participation; strengthen the screening of those with physical disabilities and limitations; and provide intervention and treatment related to cognitive improvement for older adults with poor functional health.

## Figures and Tables

**Figure 1 ijerph-19-16123-f001:**
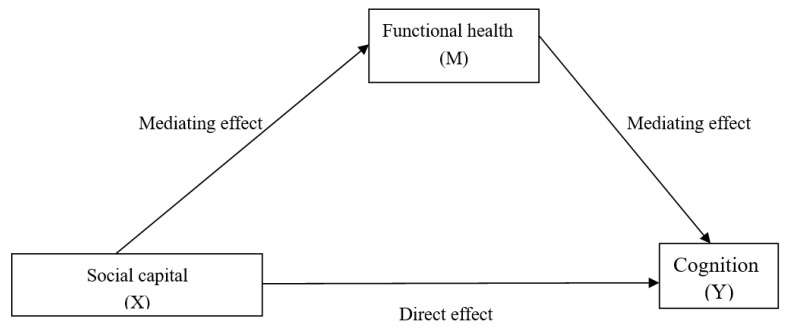
The conceptual framework of mediation analysis.

**Figure 2 ijerph-19-16123-f002:**
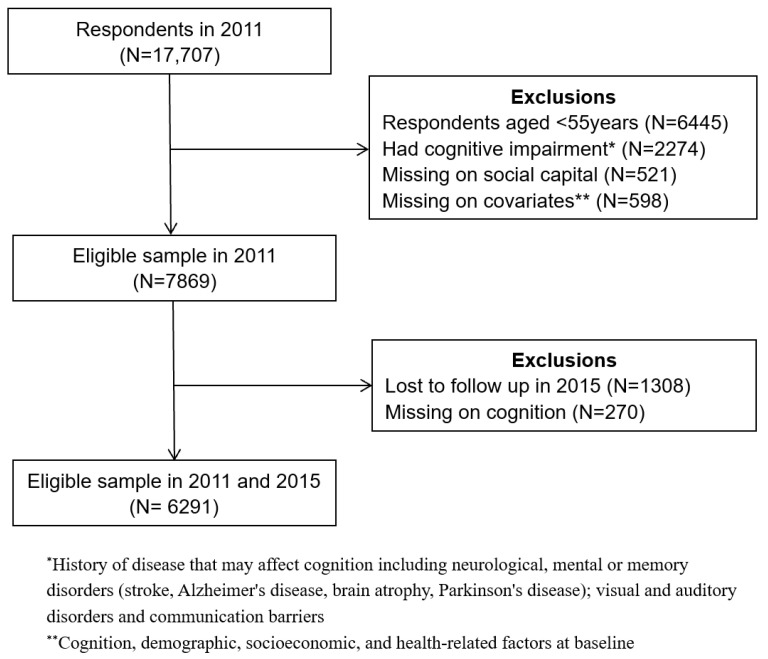
Study sample inclusion process.

**Figure 3 ijerph-19-16123-f003:**
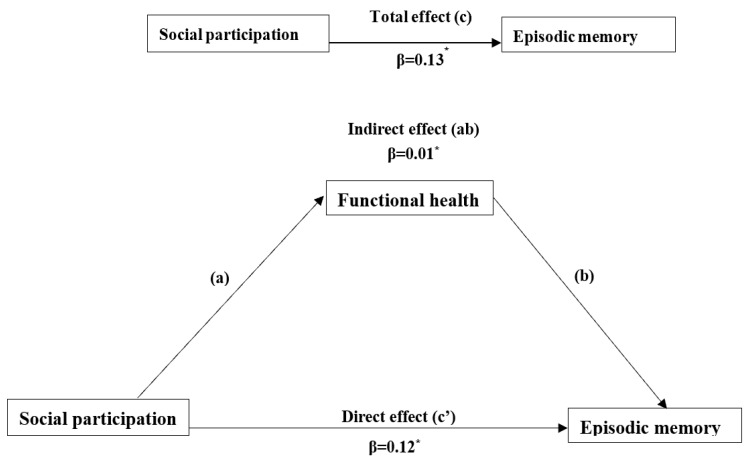
Model of the mediating role of functional health between social participation at baseline and cognition at follow-up. * *p* < 0.05. Model control for potential confounders (age, gender, marriage, money in the bank, education, hukou, self-rated health, sleeping) at baseline. c = c’ + ab.

**Table 1 ijerph-19-16123-t001:** Demographic characteristics of older adults (55+) and cognitive scores (N = 6291).

Characteristic	Total	Cognitive Score in 2015			
		Mental State		Episodic Memory	
**Total**	6291 (100.0)	6.6 (3.3)	*p*-value	3.4 (1.92)	*p*-value
**Age**			<0.001		<0.001
55–59	2261 (35.9)	7.0 (3.2)		3.8 (1.8)	
60–74	3579 (56.9)	6.6 (3.3)		3.3 (1.9)	
≥75	451 (7.2)	4.3 (3.3)		1.9 (1.9)	
**Gender**			<0.001		<0.001
Men	3053 (48.5)	7.6 (0.1)		3.5 (**0.03**)	
Women	3238 (51.5)	5.6 (0.1)		3.2 (**0.03**)	
**Marriage**			<0.001		<0.001
Married	5410 (86.0)	6.7 (3.3)		3.5 (1.9)	
Divorced	71 (1.1)	6.1 (3.3)		3.8 (2.2)	
Widowed	760 (12.1)	5.4 (3.5)		2.7 (1.9)	
Unmarried	50 (0.8)	5.8 (3.3)		3.2 (2.0)	
**Money in the bank**			<0.001		<0.001
Without	5242 (83.3)	6.3 (0.1)		3.3 (**0.03**)	
With	1049 (16.7)	7.9 (0.1)		3.8 (0.1)	
**Education**			<0.001		<0.001
Illiterate	1980 (31.5)	3.9 (2.8)		2.4 (1.9)	
Primary school or lower	2805 (44.6)	7.3 (2.8)		3.4 (1.8)	
Secondary education level	1398 (22.2)	8.8 (2.3)		4.4 (1.7)	
Higher education level	108 (1.7)	9.3 (2.0)		5.1 (1.7)	
** *Hukou* **			<0.001		<0.001
Rural *hukou*	4955 (78.8)	6.1 (3.3)		3.1 (1.9)	
Non-rural *hukou*	1292 (20.5)	8.3 (2.7)		4.3 (1.9)	
Unified *hukou*	44 (0.7)	7.6 (2.7)		3.7 (2.0)	
**Self-rated health**			<0.001		<0.001
Good	1429 (22.7)	7.2 (3.1)		3.7 (1.9)	
Fair	3258 (51.8)	6.7 (3.3)		3.4 (1.9)	
Bad	1604 (25.5)	5.7 (3.4)		3.0 (1.9)	
**Functional health**			<0.001		<0.001
unhealthy function	1726 (27.4)	5.6 (0.1)		2.9 (**0.04**)	
healthy function	4565 (72.6)	6.9 (0.1)		3.5 (**0.03**)	
**Sleep**			<0.001		<0.001
≤5	1969 (31.3)	6.0 (3.3)		3.1 (1.9)	
6	1310 (20.8)	7.0 (3.2)		3.6 (1.9)	
7	1159 (18.4)	7.1 (3.2)		3.7 (1.9)	
8	1323 (21.0)	6.8 (3.3)		3.4 (1.9)	
9	252 (4.0)	6.2 (3.3)		3.2 (2.0)	
≥10	278 (4.4)	5.5 (3.5)		2.9 (2.1)	

Note: Bold means rounding to 0.

**Table 2 ijerph-19-16123-t002:** The relationship between social capital and mental as mediated by functional health, adjusted results (N = 6291).

Social Capital Dimensions		β	SE	Z	*p*
	Effect				
High social trust	Total effect	**−0.001**	0.07	−0.01	0.992
	Direct effect	**−0.001**	0.07	−0.01	0.988
	Indirect effect	**0.0003**	**0.002**	0.15	0.878
With financial support	Total effect	0.19	0.07	2.63	0.009 *
	Direct effect	0.19	0.07	2.63	0.008 *
	Indirect effect	**−0.0003**	**0.001**	−0.20	0.842
With care	Total effect	−0.06	0.07	−0.93	0.353
	Direct effect	−0.06	0.07	−0.94	0.347
	Indirect effect	**0.001**	**0.004**	0.21	0.832
With social participation	Total effect	0.11	0.06	1.83	0.067
	Direct effect	0.11	0.06	1.82	0.068
	Indirect effect	**0.0002**	**0.003**	0.07	0.941
With reciprocity	Total effect	0.31	0.12	2.51	0.012 *
	Direct effect	0.31	0.12	2.51	0.012 *
	Indirect effect	**−0.001**	**0.003**	−0.21	0.837

Note: * = significant at *p* < 0.05; SE, standard error; with low social trust/without financial support/without care/without social participation/without reciprocity serving as the reference; bold means rounding to 0.

**Table 3 ijerph-19-16123-t003:** The relationship between social capital and episodic memory as mediated by functional health, adjusted results (N = 6291).

Social Capital Dimensions		β	SE	Z	*p*
	Effect				
High social trust	Total effect	−0.06	0.05	−1.26	0.206
	Direct effect	−0.07	0.05	−1.34	0.180
	Indirect effect	**0.003**	**0.002**	1.71	0.087
With financial support	Total effect	−0.05	0.05	−1.06	0.288
	Direct effect	−0.05	0.05	−1.02	0.308
	Indirect effect	**−0.002**	**0.002**	−1.07	0.283
With care	Total effect	−0.02	0.05	−0.33	0.741
	Direct effect	−0.03	0.05	−0.54	0.589
	Indirect effect	0.01	**0.004**	2.62	0.009
With social participation	Total effect	0.13	0.04	3.05	0.002 *
	Direct effect	0.12	0.04	2.90	0.004 *
	Indirect effect	0.01	**0.003**	2.30	0.021 *
With reciprocity	Total effect	0.08	0.09	0.99	0.323
	Direct effect	0.09	0.09	1.05	0.294
	Indirect effect	−0.01	**0.004**	−1.42	0.156

Notes: * = significant at *p* < 0.05; SE, standard error; with low social trust/without financial support/without care/without social participation/without reciprocity serving as the reference; bold means rounding to 0.

## Data Availability

The data that support the findings of this study are openly available in CHARLS at http://charls.pku.edu.cn/.

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
