# Peer review of "The Mediating Effect of Functional Health on the Relationship between Social Capital and Cognition among Chinese Older Adults"

_ijerph, 2022, doi:10.3390/ijerph192316123_

Round 1

Reviewer 1 Report

Thank you for giving me the opportunity to review the manuscript. It is interesting to note the relationship between social capital and cognition, which has the potential for early intervention to prevent cognitive decline. However, many of the previous studies referenced in this manuscript were conducted in China. I did not read the referenced articles. As this is an International Journal, please provide the data of other countries. I also have a few comments.

Lines 75-77

Please provide the citations of studies conducted in other countries.

Lines 78-89

The effects of social participation on cognitive status are inconsistent among previous studies. What reasons do the authors discuss? Please clarify.

Line 241

In this study, ADL and IADL were integrated into one variable. I wonder if basic ADL, such as dressing and bathing, were not declined in the participants of this study due to the exclusion of participants with suspected cognitive impairment. An analysis should be performed that separates the ADL and IADL.

Line 368

Why was social participation not associated with mental status when financial support and reciprocity were associated? What are the differences in the results? This may be associated with the differences in the characteristics of the tasks to evaluate mental state and episodic memory. How does the mental state evaluate the aspects of cognitive function? This should be explained and discussed in the Procedures and Discussion sections.

Lines 387-389

‘The method of measuring social trust in the study might be a possible explanation for the non-significant association between social trust and cognition [31].’

I did not understand this sentence. Please rephrase it.

Author Response

Thank you for your kind and positive comments, we have written the reply to your comments in the reply letter, please check.

Reviewer 2 Report

This manuscript considers the relationship of social capital and cognition in older adults. This is an important potential example of the social capital and health links. Social participation has many potential benefits, including in cognition, specifically for both episodic memory and mental state, and thus needs to be further studied to understand potential causal pathways for improved outcomes.

This paper uses data from a large longitudinal study, the China Health and Retirement Longitudinal Study, to investigate the relationship of social capital and cognition, specifically investigating the mediating effect of functional health (including physical activity).

The paper’s approach, methods, and results appear robust and well-written. The data source and methodology appears of high quality. In terms of analysis, the parsimonious framework presented in Figure 1 represents a valid and useful analytical design. As with much social capital research, the authors define the dimensions of social capital in the context of their study, here referencing Putnam four dimensions of (social trust, social support, social participation, and reciprocity). The authors do not attempt to evaluate the relationship of these variables. The author’s choice of proxy for social trust is whether the respondent lives in the location of their birthplace. While this may indeed predict social trust, it is deeply endogenous with other social and health characteristics. This could certainly play a role in the insignificance of the relationship for social trust as measured in this analysis.

The paper’s demonstration of a role for functional health as a mediating factor in social capital and cognition is a useful contribution, even given the small (7%) magnitude.

Specific comments

-Some editorial items (spacing, capitalization, word choice e.g. “dig out”)

-Apparently missing references in sentences of background section where there is a space before the period.

-Invalid citations

Author Response

(The authors gave the same response as above.)

Reviewer 3 Report

The topic is an essential topic about older adults’ social capital and cognition and its longitudinal data would offer a causal relationship between them. People physically live way longer, but not much research focuses on how to live, and this study will fill the avoid between social capital and cognition among older adults. If the authors revise the following, the quality of the paper would be better.  

1.     The paper should be thoroughly proofread by a professional. Some sentences are not clear enough. For example, these sentences are from the abstract “Older adults with higher 24 levels of financial support (β=0.186, P=0.009) and reciprocity (β=0.306, P=0.012) had higher mental 25 state scores Those with higher social participation levels scored higher in episodic memory (β=0.129, 26 P=0.002).” The first sentence could be rewritten as “There was a positive relationship between financial support and mental state among older adults.” “Older” and four times “higher” made not to follow the meaning quickly. Another example is “Further 351 mediation analysis found that functional health could not mediate the association be-352 tween financial support or reciprocity and mental state”on page 11. In this sentence, “could” should be “did.” I would not point out all the unclear sentences here, but I found a few examples to make my point.       

2.     Some sentences need their sources. For instance, the definitions of the key concepts stated on page 2: “Crystallised intelligence usually refers to the knowledge and skills accumulated by individuals through education or training, and fluid intelligence usually refers to the performance of individuals in learning and processing new information.” I guess the authors did not define them, so they need to provide the sources. Another statement I would like to point out is, “Similar phenomena have also been observed in many other countries, such as Canada, Austria, Germany, Japan, the Netherlands, Korea, Finland, Sweden, the U.K., and the United States” on page 2. I would not list all the sentences which need proper sources, but the authors should check them throughout the manuscript. Especially, the measurement section need more clarification. It is unclear whether the study used existing Cognition and functional health scales.

3.     It is odd to include the first name of a scholar in an academic journal unless it is an important quote. Remove “Robert” on pages 2 and 5.

4.     The discussion is nothing new. The authors should discuss the implications of the study more actively based on the existing literature and provide the worth of the study. It would be great if the authors provided any meaningful knowledge about why this study is vital for researchers, policymakers, and health professionals of older adults.      

Author Response

(The authors gave the same response as above.)

Round 2

Reviewer 1 Report

Thank you for addressing all  comments. I think the paper has been the high quality by the work you have done. 

Author Response

Once again, we would like to express our heartfelt thanks for your guidance and suggestions for our research. We are very happy that the changes we made to your suggestions have been answered to your satisfaction. We will continue to work hard and pay attention to the various problems you mentioned in our future research to improve the quality of our research. I wish you all the best!

Reviewer 3 Report

The paper should be thoroughly proofread by a professional editor. Although the author mentioned that it was carefully read by some researchers, the revision was made only by the errors I pointed out in the previous review. As I mentioned before, I cannot point out all the errors because it is not a reviewer's job. MS word and open Grammarly are not acceptable. The authors should pay for the quality of the writing. Please hire a professional English editor.   

Author Response

Thanks for pointing out!

We have invited a professional editor to proofread this paper to improve the manuscript on language, grammar, and readability thoroughly according to your opinions.

 In addition, our last round language modification was not only based on the errors you pointed out in the previous comment. We made full text language modifications, but unfortunately, there were too many modifications, and we did not retain any trace. We also apologize for the inconvenience and misunderstanding caused by not keeping the trace.

In this revision, Any revisions made to the manuscript was marked up using the “Track Changes” function.

We hope it works.
